# Tomato Maturity Classification Based on SE-YOLOv3-MobileNetV1 Network under Nature Greenhouse Environment

Fei Su [1], Yanping Zhao [1], Guanghui Wang [1], Pingzeng Liu [2,3], Yinfa Yan [1] and Linlu Zu [1,3,*]

[1] College of Mechanical and Electronic Engineering, Shandong Agricultural University, Taian 271018, China; sufei@sdau.edu.cn (F.S.); 2021110434@sdau.edu.cn (Y.Z.); 18263424609@163.com (G.W.); yanyinfa@sdau.edu.cn (Y.Y.)

[2] School of Information Science and Engineering, Shandong Agricultural University, Taian 271018, China; pzliu@sdau.edu.cn

[3] Key Laboratory of Huang-Huai-Hai Smart Agricultural Technology, Ministry of Agriculture and Rural Affairs, Taian 271018, China

[*] Correspondence: zulinlu@sdau.edu.cn

**Abstract:** The maturity level of tomato is a key factor of tomato picking, which directly determines the transportation distance, storage time, and market freshness of postharvest tomato. In view of the lack of studies on tomato maturity classification under nature greenhouse environment, this paper proposes a SE-YOLOv3-MobileNetV1 network to classify four kinds of tomato maturity. The proposed maturity classification model is improved in terms of speed and accuracy: (1) Speed: Depthwise separable convolution is used. (2) Accuracy: Mosaic data augmentation, K-means clustering algorithm, and the Squeeze-and-Excitation attention mechanism module are used. To verify the detection performance, the proposed model is compared with the current mainstream models, such as YOLOv3, YOLOv3-MobileNetV1, and YOLOv5 in terms of accuracy and speed. The SE-YOLOv3-MobileNetV1 model is able to distinguish tomatoes in four kinds of maturity, the mean average precision value of tomato reaches 97.5%. The detection speed of the proposed model is 278.6 and 236.8 ms faster than the YOLOv3 and YOLOv5 model. In addition, the proposed model is considerably lighter than YOLOv3 and YOLOv5, which meets the need of embedded development, and provides a reference for tomato maturity classification of tomato harvesting robot.

**Keywords:** tomato maturity classification; greenhouse environment; accuracy; speed; SE-YOLOv3-MobileNetV1 model



## 1. Introduction

With the development of economy and the higher level of consumption, the quality of fruits and vegetables is gradually improving. As one of the most popular fruits, the market demand for tomato is increasing, thus the greenhouse tomato planting area is also increasing [1]. Tomatoes have a high yield and a short ripening period. If ripe tomatoes are not picked in time, they will rot and deteriorate, bringing economic losses. However, if green tomatoes are picked, eating alkaloids can cause poisoning, thus the classification of tomato ripeness is very crucial [2,3]. Tomato harvesting is a time-consuming and laborsome task, especially during the picking period [4]. However, with the aging of population and the increasingly serious labor shortage [5], automatic tomato picking becomes necessary [6–8]. Accurately locating tomatoes and distinguishing their ripeness are important steps of tomato picking robots. In nature greenhouse environment, the mutual shading of branches and fruits, fluctuating illumination, complex background, etc. make the accurate identification and localization of tomatoes very difficult [9].

In current studies, hand-crafted features combined with machine vision algorithm are explored to detect and locate tomatoes automatically. Zhao et al. extracted the Haar-like features to train AdaBoost classifier, then combined them with color analysis based on

the average pixel value to detect ripe tomatoes. However, the detection speed is low and cannot be applied to more varied unstructured environments [10]. Liu et al. used the Histograms of oriented gradients (HOG) descriptor to train the support vector machine (SVM) classifier. Herein, a coarse-to-fine scanning method was proposed to detect mature tomatoes. Although F1 reaches 90.00%, this method is not satisfactory for the overlapped and occluded tomatoes, and not only ripened ones [11]. To solve the problem of tomato fruit, the overlap caused by detection accuracy decreased. Chen et al. used machine vision to detect and locate ripe fruits, including image graying, binary image, and other methods. Through experiments on tomatoes and citrus, the detection effect of ripe fruits and fruits in the near zone is very impressive. However, the influence of occlusion and light on detection in nature environment is solved only to a certain extent [12].

With the development of deep learning algorithm based on convolutional neural network (CNN) [13], problems with fruit overlap and illumination variation will be solved [14]. Liu et al. proposed an improved DenseNet network to detect ripe tomatoes with an accuracy of 91.26% under illumination interference, but the model performance was bad when detecting tomatoes with different sizes and similar color interferences [15]. Xu et al. improved YOLOv3-tiny model to detect tomatoes, where the improved depth-separable convolution and a residual structure were used to reduce the number of FLOPs, thus the speed of this model was fast [16]. Tomato are climacteric fruits, one must considerably account for transportation and storage time for an optimal harvest. For example, tomatoes can be harvested in the physiological maturity stage (green), which can complete the ripening after harvest. Therefore, it is necessary to accurately locate and distinguish the maturity of tomatoes for different harvesting purposes [17–19]. Zhang et al. improved the deep learning-based classification method to classify tomato maturity by inserting two layers of max-pooling in CNN layers. This method has less parameter calculation and higher accuracy [20]. Wan et al. detected the maturity levels of tomatoes by combining the feature color value and backpropagation neural network classification technology in the laboratory environment [21]. However, the classification process should be carried out in nature environment rather than laboratory environment.

Considering the above studies, the speed and accuracy of tomato maturity classification are two major problems that limit automatic tomato picking. In terms of speed, maturity classification is mostly carried out in laboratory environment [21], which is relatively simple and lacks discussion in nature environment, thus failing to meet the requirements of real-time detection of tomato maturity classification [21,22]. In terms of accuracy, there is a lack of research on tomato maturity classification, and the picking period cannot be accurately determined, thus failing to meet the requirements of tomato maturity accuracy. The main contributions of this paper are as follows:

1.  The lightweight SE-YOLOv3-MobileNetV1 network is proposed to distinguish four types of maturity: <60%, 61%_70%, 71%_80%, and 81%_100% [23] under nature greenhouse environment.
2.  The backbone network of YOLOv3 is replaced with MobileNetV1 [24] to improve the speed of network detection.
3.  The Squeeze-and-Excitation [25] attention mechanism module, mosaic data augmentation, and K-means clustering algorithm are used to improve the classification accuracy of tomato maturity.

The model proposed in this paper can be used for precise positioning and maturity level classification, with high detection speed, which lays a foundation for the tomato harvesting robot. The process of the paper is as follows: Section 2 introduces the creation of tomato data and the method of tomato maturity detection. In Section 3, the performance of tomato maturity detection model is analyzed and discussed. Finally, conclusions are drawn in Section 4.

## 2. Materials and Methods

*2.1. Data Acquisition and Annotation*

The data used in the experiment were collected in the Tomato Solar Greenhouse, Science and Technology Innovation Park of Shandong Agricultural University, Tai'an City, China. The variety of tomato used in this experiment was "Sheng Luo Lan 3689". The capture device was vivo S5 rear camera, 48 megapixel main camera +800, the resolution of the original image was 4000 × 3000 pixels, then adjusted to 800 × 600 pixels. The images were captured at different distances, angles, lighting conditions, and time periods. For example, Figure 1 includes tomato images without occlusion and overlap, heavily overlapped tomato images, branches occlusion tomato images, leaves occlusion tomato images, branches and leaves mixed occlusion tomato images, and slightly overlapped tomato images. The maturity index is a key to determine the harvest time of tomato, and the color of a tomato is the most significant indicator of maturity [26]. Therefore, in this paper, tomato maturity is divided into four grades: <60%, 61%_70%, 71%_80% and 81%_100%, which are shown in Figure 2.

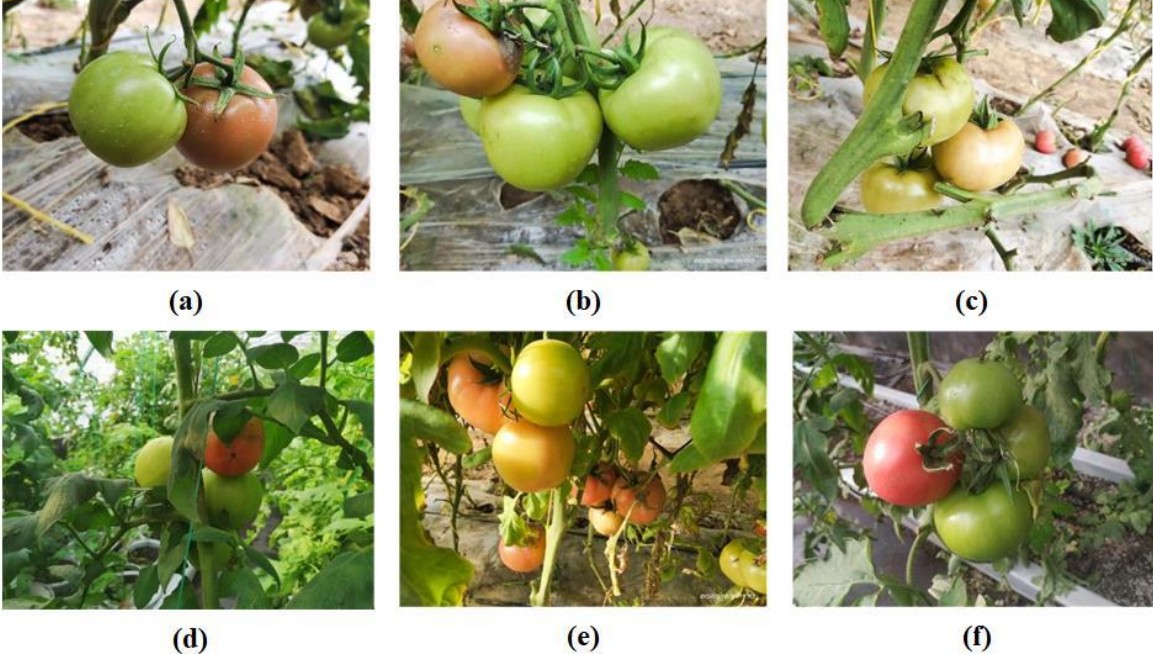

**Figure 1.** Display of partial collected images. (**a**) No occlusion and no overlap tomato image; (**b**) heavily overlapped tomato images; (**c**) branches occlusion tomato images; (**d**) leaves occlusion tomato images; (**e**) branches and leaves mixed occlusion tomato images; (**f**) slightly overlapped tomato images.

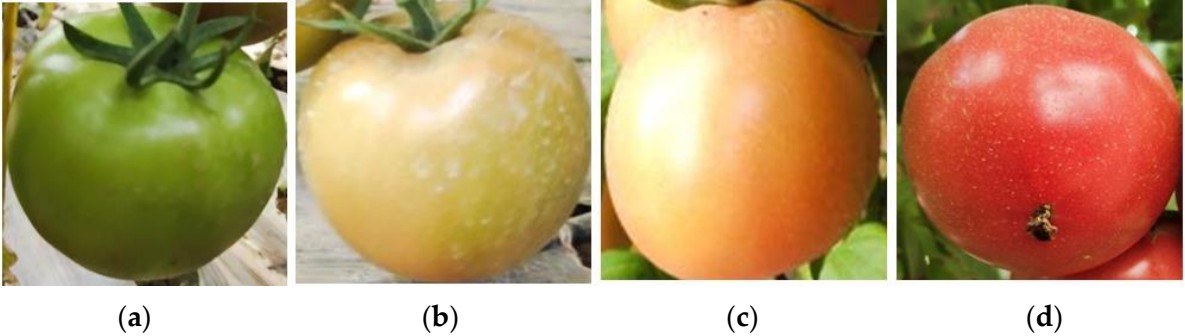

**Figure 2.** Images of four different ripening tomatoes. (**a**) <60%; (**b**) 61%_70%; (**c**) 71%_80%; (**d**) 81%_100%.

Through manual screening, 462 original tomato images were selected, then LabelImg was used for manual annotation. The data set is divided into training set and test set in a ratio of 8:2, i.e., the training set has 369 images and the test set has 93 images. To avoid network overfitting and solve the problem of unbalanced data samples, the number of training set is expanded to 1476 through horizontal flip, vertical flip, and horizontal vertical flip [27]. A total number of 5147 tomatoes are included in the data set, and the allocation of the number of different types of tomatoes in the training set and test set is shown in Figure 3. Meanwhile, mosaic data enhancement is used in the training process to randomly combine four images into one image, which can enrich the background of the detection target and improve the detection effect of small targets.

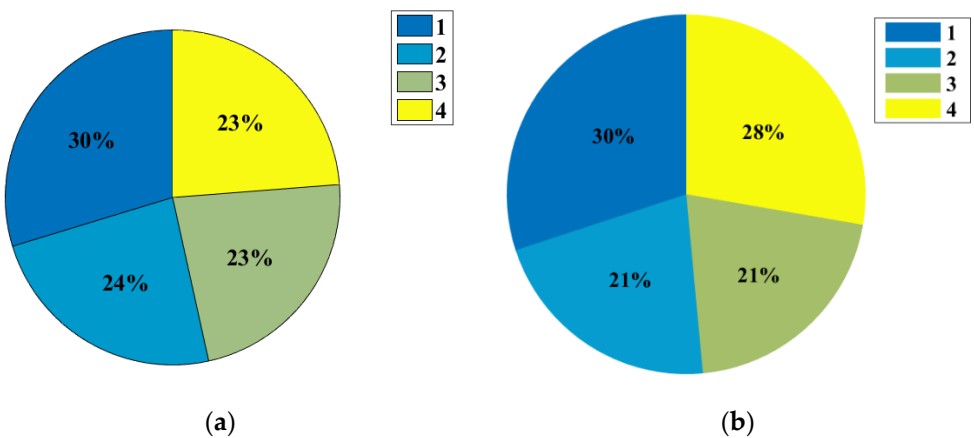

(**a**)  (**b**)

**Figure 3.** Number of tomatoes in the training set and test set. (**a**) Training set; (**b**) test set. The figures in the legend represent 1: <60%, 2: 61%_70%, 3: 71%_80%, 4: 81%_100% of the maturity of the four grades of tomato in the percentage of the data set.

### 2.2. Tomato Maturity Classification Network Selection and Improvement

### 2.2.1. Network Selection

In the actual situation, the camera shooting angle and distance are random, which will make the volume of tomato in the collected plane image uneven. If the tomato feature map of a single scale is used for detection, the detection accuracy of small tomato is not high. Coincidentally, YOLOv3 adopts multi-scale prediction method to solve the problem of inaccurate detection of small targets to a certain extent. The low-level feature map is processed less times by the network, thus it contains more target location information, but the feature information is not clear. The high-level feature map after multi-layer convolutional operation of the network contains more feature information, but the target location information is less. YOLOv3 makes them complement each other by means of feature fusion at different scales, and performs multi-scale detection on targets by combining feature map information at multidimensional scales. YOLOv3 uses feature map information of small scale, medium scale, and large scale to conduct feature fusion through convolution kernel, and independently predicts these three dimensions, obtaining three prediction results. Therefore, this paper selects YOLOv3 as the basic network. However, to meet the requirements of tomato harvesting robot, the detection speed and accuracy need to be improved.

### 2.2.2. Network Speed Improvement

The original backbone network of YOLOv3 is DarkNet53, which has deep layers and complex structure. To improve the detection speed, the backbone network of YOLOv3 is replaced by lightweight MobileNetV1, which greatly reduces the number of parameters and accelerates the model. The main concept of MobileNetV1 is to use depthwise separable convolution, which divides standard convolution into depthwise convolution and

pointwise convolution [23]. The comparison between standard convolution and depthwise separable convolution is shown in Figure 4.

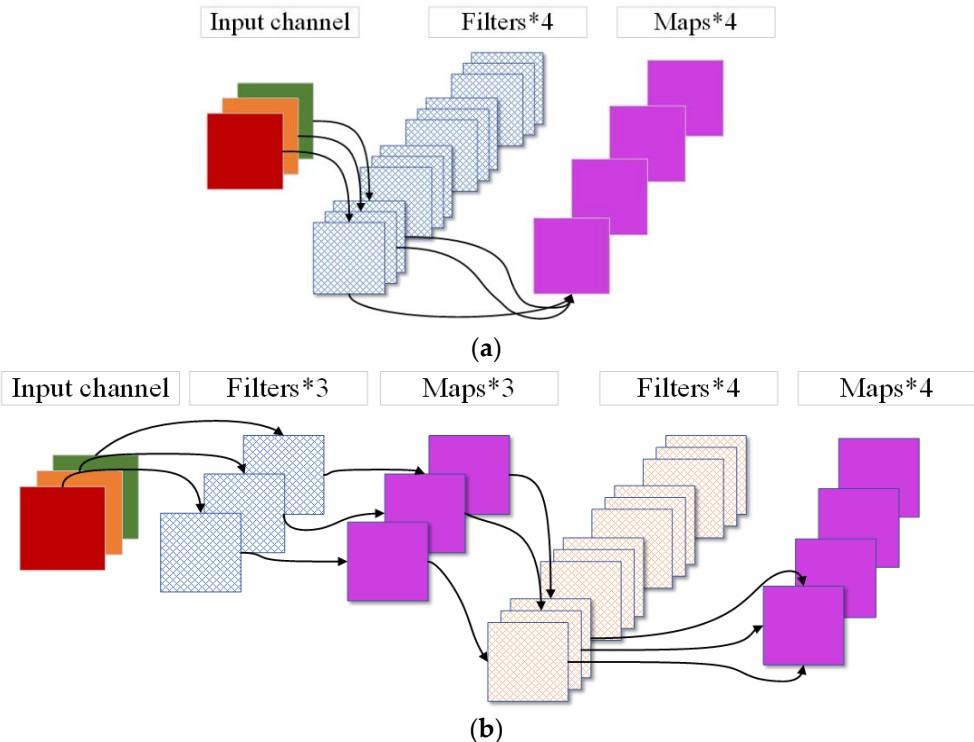

(a)

(b)

**Figure 4.** Comparison between standard convolution (**a**) and depthwise separable convolution (**b**). Here Filters*3 indicates that there are three Filters, and so on.

One convolution kernel in standard convolution acts on all channels of the input feature map, the depth of each convolution kernel corresponds to the number of channels of the input, and the number of the convolution kernel corresponds to the number of the output feature map. However, each convolution kernel in the depthwise convolution part of depthwise separable convolution only acts on one channel of the input. The pointwise convolution process is similar to the standard convolution process, and its main function is to combine the convolution results of the previous layer. If the size of the input feature map is $D_F \times D_F \times M$, the size of the convolution kernel is $D_K \times D_K$, and the size of the output feature map is $D_F \times D_F \times N$, the following equations can be obtained:

$$\text{Standard convolution} : \ D_K \times D_K \times M \times N \times D_F \times D_F \tag{1}$$

$$\text{Depthwise separable convolution} : \ D_K \times D_K \times M \times D_F \times D_F + M \times N \times D_F \times D_F \tag{2}$$

It can be seen from Equation (2) that the computation amount of deeply separable convolution is the sum of the computation amount of deep convolution and point-by-point convolution. According to Equation $\frac{(2)}{(1)} = \frac{1}{N} + \frac{1}{D_K^2}$, depthwise separable convolution greatly reduces the computation amount compared with standard convolution.

### 2.2.3. Network Accuracy Improvement

After speed improvement, the improved network performs well on detection speed. However, the detection accuracy has been reduced due to the reduction of the calculation parameters. A Squeeze-and-Excitation (SE) module is added at the end of network feature extraction layer and in front of each detection layer to greatly improve detection accuracy and maintain speed. Due to the added SE module, this could lead the network to allocate different attentions to different channel features; namely, to assign a large weight to impor-

tant feature channels and a small weight to irrelevant feature channels, to ignore useless information and improve detection accuracy.

The SE steps are illustrated in Figure 5, which contains both Squeeze and Excitation. The Squeeze operation is to compress the W∗H∗C eigenvectors into 1∗1∗C through global average pooling. The Excitation operation is to add the eigenvectors of 1∗1∗C to the full connection layer to predict the importance of channels, and then apply the importance of different channels to the corresponding channels of the previous feature map. The scale operation is to multiply the weight value of each channel obtained by excitation operation with the weight of the corresponding channel in the original feature map, and finally the weighted feature map is used as the input of the next layer network.

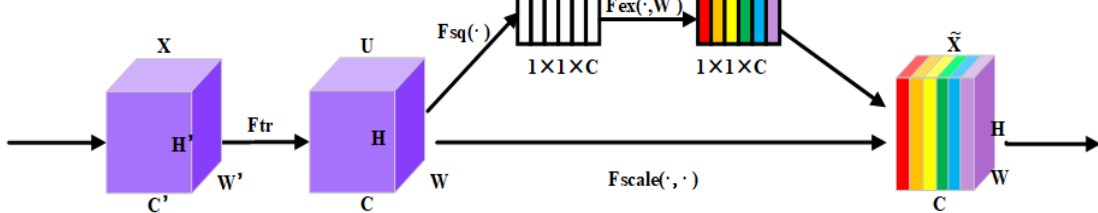

**Figure 5.** Details of the Squeeze-and-Excitation operation. H′, W′, C′, H, W, and C are the height, width, and channel number of feature map X, U, and $\widetilde{X}$, respectively, $F_{tr}$ represents traditional convolution operation, $F_{sq}(\cdot)$ represents squeezing operation, $F_{ex}(\cdot, W)$ represents excitation operation, $F_{scale}(\cdot, \cdot)$ represents scale operation.

The improved YOLOv3 algorithm is called SE-YOLOv3-MobileNetV1, and the corresponding network structure diagram of the model is shown in Figure 6. The anchor box size in the original YOLOv3 network configuration file is taken from the Common Objects in Context (COCO) data set. If the tomato image data set is directly trained with anchor box parameters, the detection result will be inaccurate. Therefore, in this paper, K-means clustering algorithm is used to recalculate the sizes of anchor boxes based on the tomato image training data set, which are suitable for small size targets (30,39), (47,57), (61,77), medium size targets (79,102), (98,126), (119,153), and large size targets (141,180), (170,216), (214,263), respectively. The model construction process is shown in Figure 7.

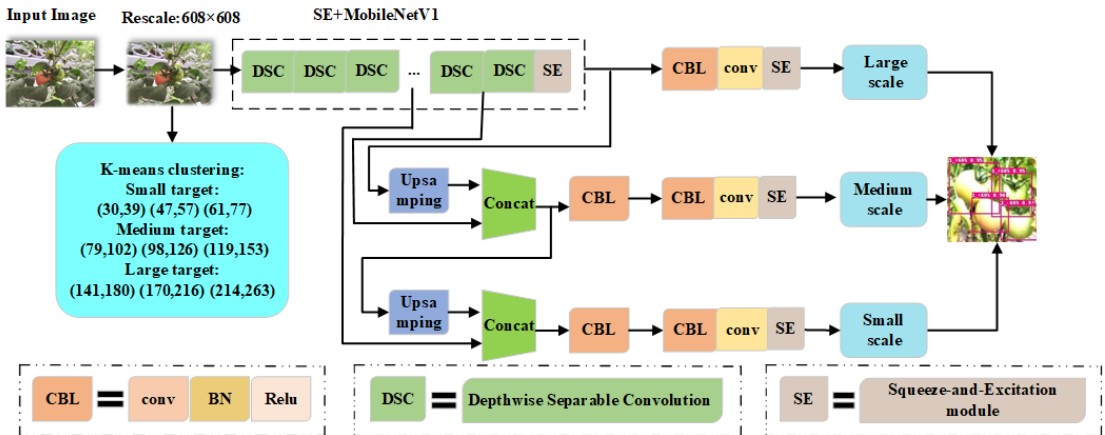

**Figure 6.** The structure diagram of SE-YOLOv3-MobileNetV1 network.

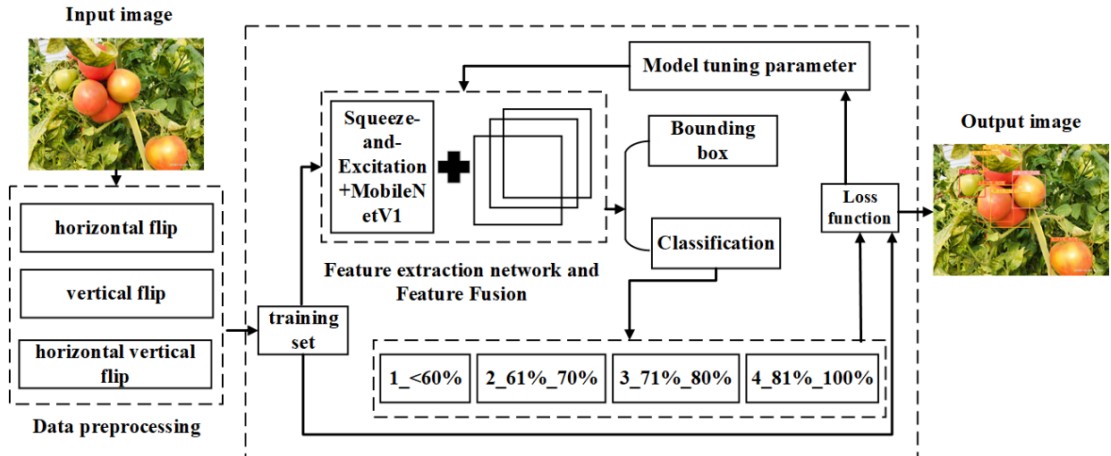

**Figure 7.** The construction process of tomato classification model.

*2.3. Loss Function Improvement*

To use YOLOv3 for end-to-end training, the loss function needs to be established, which is an important reference index to measure whether the model converges successfully. The value of the loss function is obtained in the forward propagation of the neural network. In the back propagation, the difference between the predicted value of the trained model and the actual value can be estimated through the gradient change of the loss value, and the training of the model can be adjusted by feedback. In general, the smaller the loss function value, the better the model performance. YOLOv3 establishes a loss function for each prediction box generated in the input image, but the specific equation in the original paper is not clearly stated [28]. Here, the loss function is expressed by Equations (3)–(6) in combination with the establishment method of loss function in YOLOv1 and YOLOv2:

$$Loss_{box} = \lambda_{coord} \sum_{i=0}^{S^2} \sum_{j=0}^{B} I_{i,j}^{obj}(2 - w_i \times h_i)[(x_i - \hat{x}_i)^2 + (y_i - \hat{y}_i)^2 + (w_i - \hat{w}_i)^2 + (h_i - \hat{h}_i)^2] \tag{3}$$

$$Loss_{obj} = -\lambda_{noobj} \sum_{i=0}^{S^2} \sum_{j=0}^{B} I_{ij}^{noobj}[\hat{C}_i \log(C_i) + (1 - \hat{C}_i) \log(1 - C_i)] - \sum_{i=0}^{S^2} \sum_{j=0}^{B} I_{ij}^{obj}[\hat{C}_i \log(C_i) + (1 - \hat{C}_i) \log(1 - C_i)] \tag{4}$$

$$Loss_{class} = -\sum_{i=0}^{S^2} I_{ij}^{obj} \sum_{c \in classes} [\hat{p}_i(c) \log(p_i(c)) + (1 - \hat{p}_i(c)) \log(1 - p_i(c))] \tag{5}$$

$$Loss_{object} = Loss_{box} + Loss_{obj} + Loss_{class} \tag{6}$$

Here, $\hat{x}_i$, $\hat{y}_i$, $\hat{w}_i$, and $\hat{h}_i$ correspond to the central coordinates and width and height of the actual box, respectively, $x_i$, $y_i$, $w_i$, and $h_i$ correspond to the central coordinates and width and height of the predicted box, respectively.

In this paper, Complete-IoU [29] is used for bounding box regression, which could take into account the distance, scale, overlap area, and aspect ratio between target and anchor compared with IoU. Therefore, regression becomes more stable and effectively avoids divergence of the training process model. The Equations are as follows:

$$L_{CIoU} = 1 - IoU + \frac{\rho^2(b, b^{gt})}{r^2} + \alpha v \tag{7}$$

$$\alpha = \frac{v}{1 - IoU + v} \tag{8}$$

$$v = \frac{4}{\pi^2}(\arctan\frac{w^{gt}}{h^{gt}} - \arctan\frac{w}{h})^2 \tag{9}$$

Here, *r* represents the diagonal distance of the smallest closure area that can contain both the prediction bounding box and the ground truth bounding box, $\rho^2(b, b^{gt})$ represents the Euclidean distance between the center point of the predicted anchor and the real anchor, and $1 - IoU$ is used to obtain the corresponding loss.

Therefore, the loss function in this paper is expressed by Equation (10) as follows:

$$
\begin{aligned}
Loss_{object} = 1 - IoU + \frac{\rho^2(b, b^{gt})}{c^2} + \alpha v - \lambda_{noobj} \sum_{i=0}^{S^2} \sum_{j=0}^{B} I_{ij}^{noobj} [\hat{C}_i \log(C_i) + (1 - \hat{C}_i) \log(1 - C_i)] - \\
\sum_{i=0}^{S^2} \sum_{j=0}^{B} I_{ij}^{obj} [\hat{C}_i \log(C_i) + (1 - \hat{C}_i) \log(1 - C_i)] - \sum_{i=0}^{S^2} I_{ij}^{obj} \sum_{c \in classes} [\hat{p}_i(c) \log(p_i(c)) + (1 - \hat{p}_i(c)) \log(1 - p_i(c))]
\end{aligned}
\tag{10}
$$

According to Equations (3)–(6) and (10), the loss function is composed of three parts: Box loss, confidence loss, and classes loss. In Equations (3)–(6) and (10), $C_i$ represents the predicted category, $\hat{C}_i$ represents the actual category, $\hat{p}_i(c)$ represents the actual value to which an actual box belongs to category *c*, $p_i(c)$ represents the probability that a predict box belongs to category c. In addition, a prediction is made with the target (obj) in the prediction box, and a prediction is made without the target (noobj); where, $\lambda_{coord}$ and $\lambda_{noobj}$ are the additional weight coefficients for coordinate prediction and target-free confidence, in which $\lambda_{coord}$ is usually 5 and $\lambda_{noobj}$ is 0.5. The three components of the loss function are shown in Table 1.

**Table 1.** YOLOv3 loss function composition.

| Index | Type of Loss | The Meaning of Loss |
|:---:|:---:|:---:|
| 1 | Box loss | Compare the position of the prediction box with the true box |
| 2 | Confidence loss | Confidence prediction of the target in the prediction box |
| | | Confidence prediction without target in the prediction box |
| 3 | Classes loss | Comparison between the predicted target category and the actual category |

*2.4. Network Visualization*

As a typical end-to-end one-stage target detection model, YOLOv3 inputs a tomato image during image analysis, and the detection result will be directly output by the network, but the study of intermediate learning process becomes difficult. The poor interpretability of the network hinders further optimization of the network, thus it is necessary to visualize the model to better understand the learning process. Generally, features contain less semantic information and more location information in shallow layers. With the number of feature layers receiving deeper, features contain more semantic information and less location information. Feature visualization, convolution kernel visualization, and class activation mapping visualization are commonly used methods of feature visualization [30,31]. In this paper, gradient-weighted class activation map (Grad-CAM) [32] is used to visualize networks. The principle is: Network receives feature map and categories of output predicted by the forward propagation, and then the gradient information of the feature map can be obtained by backward propagation of the predicted output value. The importance of each channel of the feature map can be obtained by taking the mean value of the gradient information on W and H. The sum of data of each channel of feature map is weighted, and finally the ReLU function is used.

**3. Results**

The training and evaluation of the model are carried out under the Windows 10 operating system, CPU Model: Intel(R) Core (TM) i7-8700 CPU @ 3.20 GHz (12CPUs), ~3.2 GHz, the memory size is 16,384 MB and python3.7.0 is used. The Pytorch framework is

used to train the network model, with a learning rate of 0.001 with momentum of 0.937 and weight decay of 0.0005.

To fully verify the effectiveness of the proposed model in detecting tomato maturity and its robustness under uneven illumination and complex environmental conditions, the same data set is used to train four models, YOLOv3, YOLOv3-MobileNetV1, SE-YOLOv3-MobileNetV1, and YOLOv5, for 200 rounds of training. In addition, early stopping techniques are used to prevent network overfitting. Then, the real-time solution of loss function and mAP can determine whether the training trend is correct to ensure the normal convergence of the model with the increase of iterations. If there is a problem in the training process, the training can be stopped in time to avoid time waste. The visual tool TensorBoard is used to record the changes of loss function value and mean average accuracy during the training process. Figure 8 shows the mAP curve images and the loss curve images of the four models. It can be seen from mAP and loss curves of the four models that the model training converges normally.

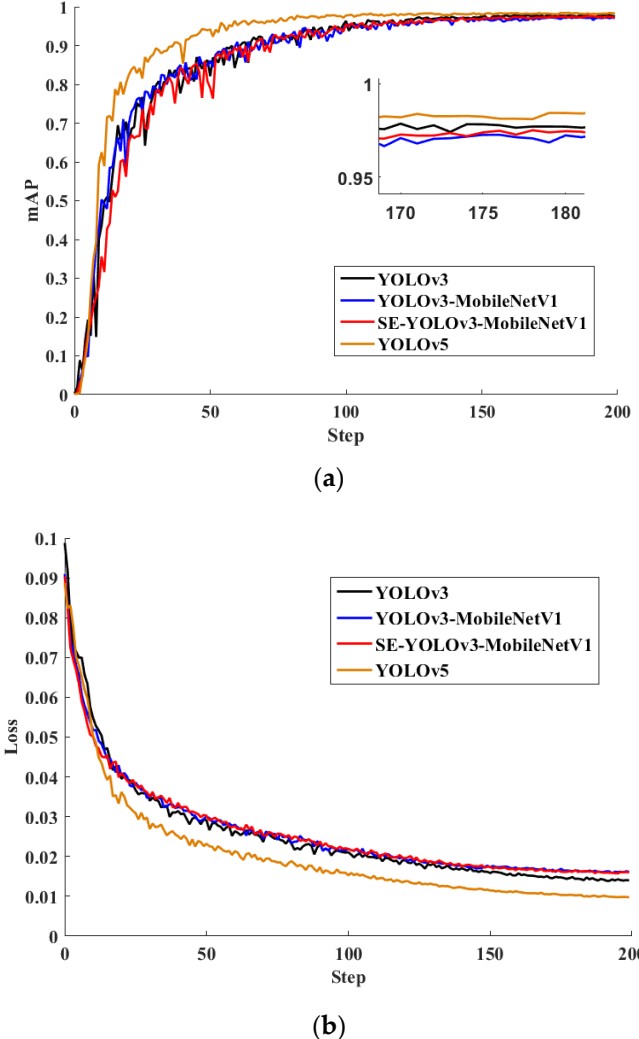

**Figure 8.** The mAP curve images (**a**) and the loss curve images (**b**) of the four models.

## 3.1. Evaluation Indices of the Model

The mean average precision (mAP), F1-score, detection speed, and model size are used to evaluate the performance of the model. Equation (11) is used to calculate mAP:

$$\text{mAP} = \frac{1}{S_{cla}} \sum\nolimits_{i \in s_{cla}} AP(i) \tag{11}$$

Equation (12) is used to calculate accuracy and harmonic mean of recall rate:

$$\text{F1} - \text{Score} = 2 \times \frac{precision \times recall}{precision + recall} \tag{12}$$

where $precision = TP/(TP+FP)$, $recall = TP/(TP+FN)$, $TP$ indicates that the target is a tomato, and is correctly detected as a tomato. $FP$ indicates that the target is not a tomato, but has been incorrectly detected as a tomato. $FN$ indicates that the target is a tomato, but has been incorrectly detected as not a tomato.

The mAP, F1-score, detection speed, and model size comparisons are shown in Table 2. YOLOv5 has the best mAP value, followed by YOLOv3. Compared with YOLOv3 and YOLOv5, the mAP of YOLOv3-MobileNetV1 is 0.7% and 1.2% lower, while the proposed SE-YOLOv3-MobileNetV1 model is only 0.2% and 0.9% lower. For model size, the YOLOv3-MobileNetV1 model is the smallest. The model size of the YOLOv3 and YOLOv5 model is 2.5 and 1.9 times than the SE-YOLOv3-MobileNetV1 model. To sum up, the SE-YOLOv3-MobileNetV1 model is more suitable for embedded devices.

**Table 2.** Performance comparison of four models.

| Model | mAP@0.5 (%) | F1-Score (%) | Model Size (MB) |
|---|---|---|---|
| YOLOv3 | 97.9 | 95.2 | 117.0 |
| YOLOv3-MobileNetV1 | 97.2 | 94.4 | 46.4 |
| YOLOv5 | 98.4 | 96.6 | 88.5 |
| SE-YOLOv3-MobileNetV1 | 97.5 | 94.9 | 46.7 |

*3.2. Performance Comparison of the Models on the Test Set*

The trained model needs to have good generalization, thus it is necessary to test the performance on test set. YOLOv3 model has the highest mAP value, but lowest detection speed. To improve the detection speed and make the network more suitable for embedded devices, the backbone network of YOLOv3 is replaced by MobileNetV1, i.e., the YOLOv3-MobileNetV1 model, which is 285.5 ms faster than YOLOv3. However, the drawback of this operation is the reduced detection accuracy, the mAP value is 1.2% smaller than YOLOv3. Therefore, the attention mechanism module is added, i.e., the proposed SE-YOLOv3-MobileNetV1 model, the mAP value is improved by 0.2% than YOLOv3-MobileNetV1 model. Due to the addition of the attention mechanism module, the number of parameters in SE-YOLOv3-MobileNetV1 model increases, and the detection speed is 6.9 ms slower than YOLOv3-MobileNetV1.

The performance of SE-YOLOv3-MobileNetV1 model is also compared with YOLOv5, the latest model of the YOLO series, with 1.7% higher mAP value and 236.8 ms faster detection speed. After the above analysis, SE-YOLOv3-MobileNetV1 model can balance detection precision and generalization performance better, especially that the detection time meets the requirements of real-time detection. The performance of YOLOv3, YOLOv3-MobileNetV1, SE-YOLOv3-MobileNetV1, and YOLOv5 models in the test set is summarized in Table 3.

**Table 3.** Performance of the six models on the test set.

| Model | mAP@0.5 (%) | Detection Speed (ms) |
|---|---|---|
| YOLOv3 | 88.7 | 505.7 |
| YOLOv3-MobileNetV1 | 87.5 | 220.2 |
| YOLOv5 | 86 | 463.9 |
| SE-YOLOv3-MobileNetV1 | 87.7 | 227.1 |

A detailed comparison of model performance on four different tomato maturity tests are provided in Figure 9. The detection accuracy of 3_71%_80% tomato of four models is the lowest mainly due to two reasons: First, tomatoes in this stage are similar to those of

the second stage in color and shape, which is easily confused in data annotation, resulting in low detection accuracy. Second, due to the small number of tomatoes in this stage, the characteristics of network learning are relatively few, which increases the detection difficulty. In addition, the detection accuracy of 1 _ < 60% tomato is not ideal, mainly due to the fact that the color of the branches and leaves of the tomatoes are very similar to the tomatoes, which pose great challenges to detection. The detection accuracy of the 4_81%_100% type is the highest, since the color and size of tomatoes in this period are very different from the background.

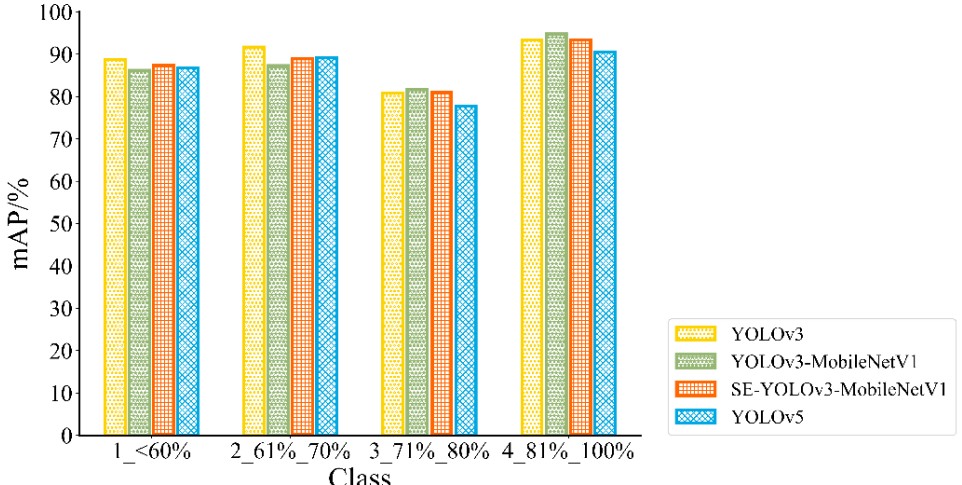

**Figure 9.** Test results of four different models on four different maturity levels.

### 3.3. Visualization of the SE-YOLOv3-MobileNetV1 and YOLOv3-MobileNetV1 Models

To better understand how the network accurately classifies the targets in the original image data, in this paper, SE-YOLOv3-MobileNetV1 and YOLOv3-MobileNetV1 models are visualized by Grad-CAM. Through visualization, it can be seen that the corresponding pixel in the input image has an impact on category classification, and a reasonable explanation of the results provided by the network can be made to see that the network performance is better with the SE module added. Figure 10a,c shows the visualization results of SE-YOLOv3-MobileNetV1 model, and Figure 10b,d shows the visualization results of YOLOv3-MobileNetV1 model. Figure 10b,d incorrectly focuses on the yellow blades and red valve features, causing the network to mistakenly identify yellow blades and red valves as 2_61%_70% and 4_81%_100% categories.

### 3.4. Analysis of Test Results

Tomato images are randomly selected for testing, and the results in Figure 11 show that the all models have good performance in tomato maturity identification and location under the condition of uneven illumination and mutual occlusion of leaves, branches, and fruits. However, for some special situations, the model proposed in this paper works better. The yellow mark is the undetected tomato, and the possible reason is that tomato targets are small, thus it is difficult to detect. Another main reason is that tomatoes in this period are close to the color of leaves and branches, thus it is easy to be missed and mis detected. The model proposed in this paper can well avoid the problem of missed detection and false detection, and at the same time, it can accurately locate tomato and distinguish the maturity of tomato.

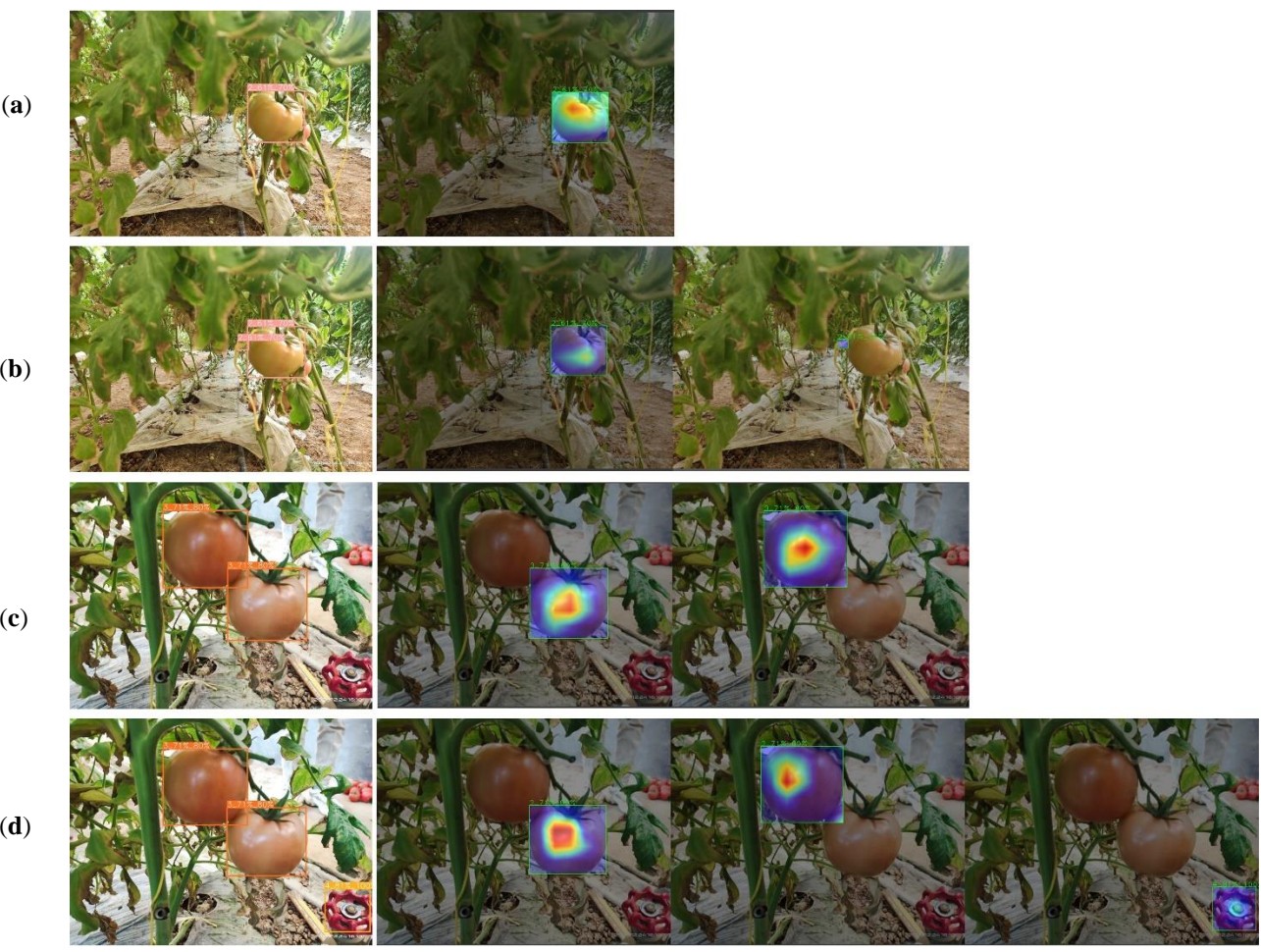

**Figure 10.** Grad-CAM visualization. (**a**) SE-YOLOv3-MobileNetV1; (**b**) YOLOv3-MobileNetV1; (**c**) SE-YOLOv3-MobileNetV1; (**d**) YOLOv3-MobileNetV1.

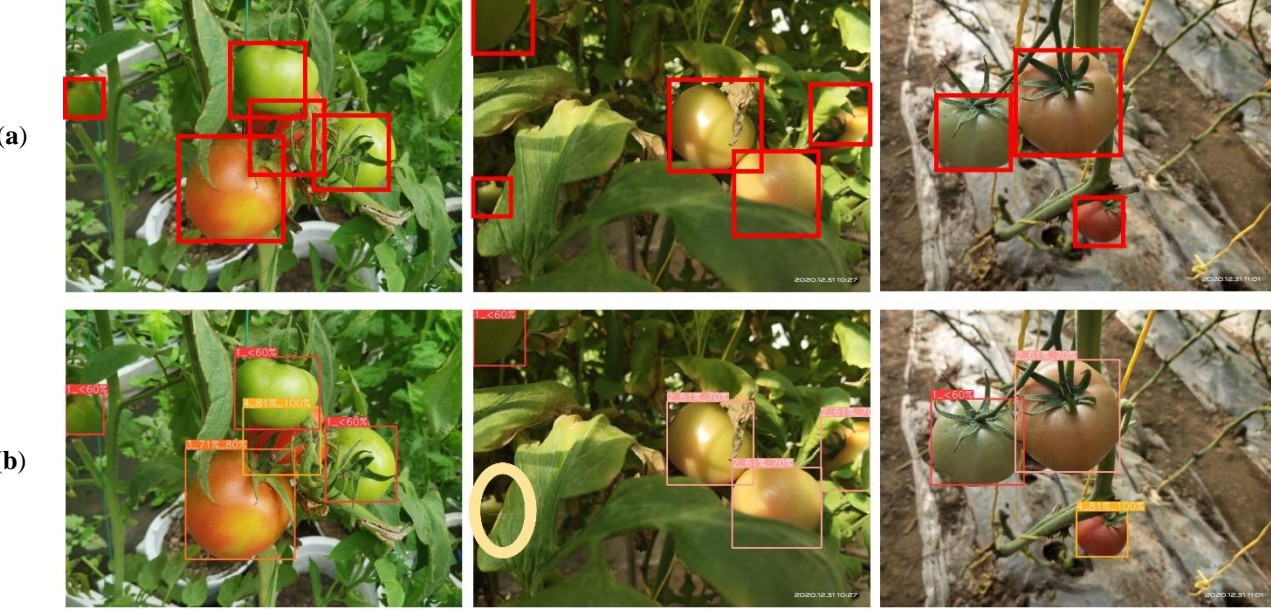

**Figure 11.** *Cont.*

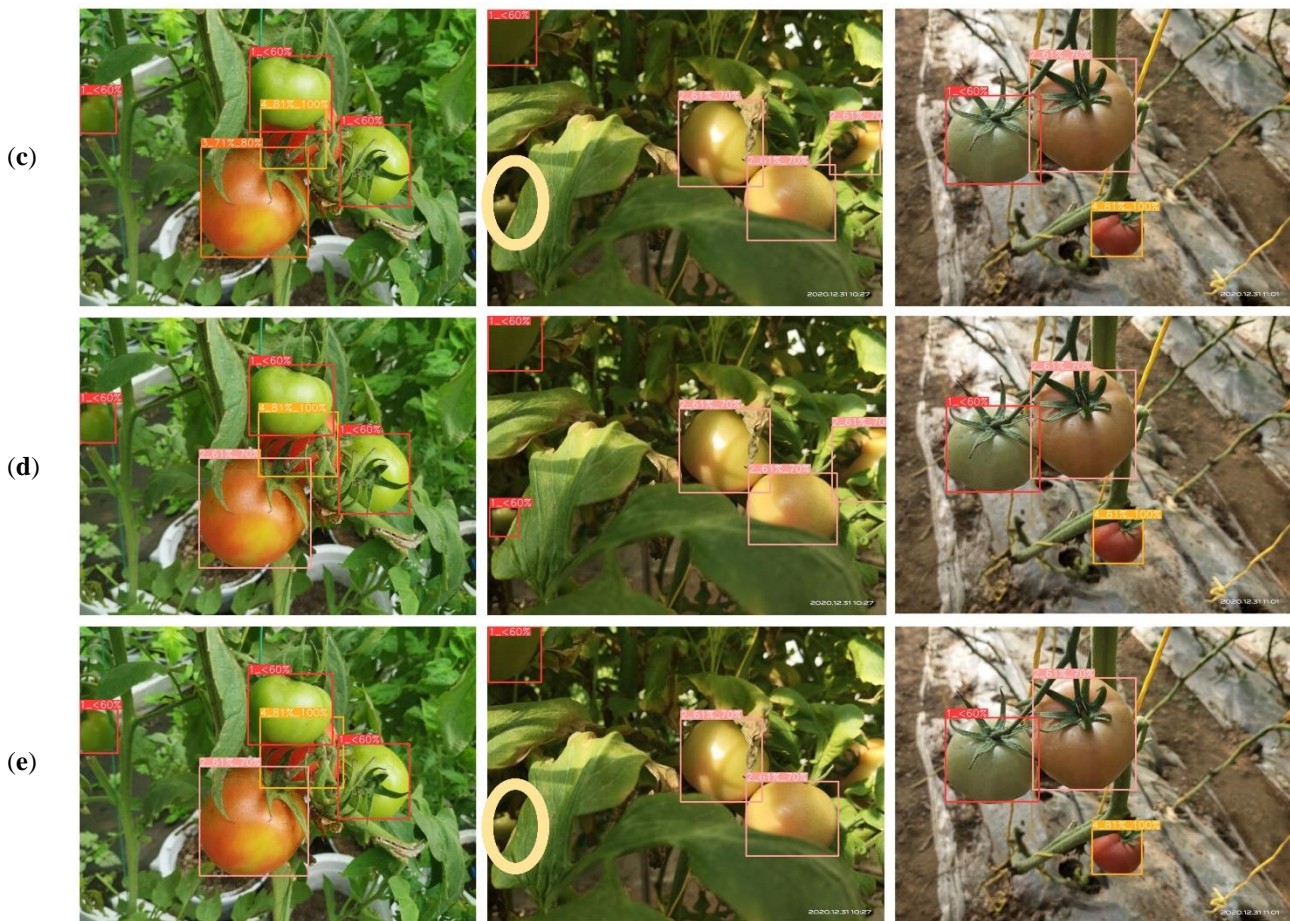

**Figure 11.** The predicted performance of four models. (**a**) Manually marked initial image; (**b**) YOLOv3 model; (**c**) YOLOv3-MobileNetV1 model; (**d**) SE-YOLOv3-MobileNetV1; (**e**) YOLOv5.

## 4. Discussion

The proposed SE-YOLOv3-MobileNetV1 model in this paper is compared with the existing tomato maturity classification methods, including laboratory environment, greenhouse environment, and the comparison of different maturity classifications. The results are shown in Table 4. Compared with traditional machine learning methods, the model in this paper has higher mAP value and can distinguish four kinds of maturity. Compared with the laboratory environment, although the mAP is relatively low, the greenhouse conditions are more in line with the requirements of tomato harvesting robot. Through comparison, it can be concluded that the SE-YOLOv3-MobileNetV1 model proposed in this paper can successfully detect four kinds of tomatoes with different maturity levels under nature greenhouse environment.

**Table 4.** The proposed method is compared with existing methods.

| Source | Method | Environment | Maturity Level | mAP |
|---|---|---|---|---|
| Zhao et al. [10] | Adaboost | Greenhouse | Ripe tomato | 96.5% |
| Liu et al. [15] | DenseNet | Various film and television works, etc. | Ripe tomato | 91.26% |
| Wan et al. [21] | BPNN | Laboratory | Three maturity levels | 99.31% |
| Chen et al. [33] | YOLOv3-DPN | Greenhouse | All tomatoes | 96.83% |
| Lawal et al. [34] | YOLO-Tomato-C | Greenhouse | Two maturity levels | 99.5% |
| Rupanagudi et al. [35] | image processing | Laboratory | Six maturity levels | >98% |
| Proposed method | SE-YOLOv3-MobileNetV1 | Greenhouse | Four maturity levels | 97.5% |

### 5. Conclusions

In this paper, a real-time tomato maturity detection model SE-YOLO-MobileNetV1 is proposed. To meet the real-time requirements of automatic tomato picking, the YOLOv3 algorithm is improved, and the main conclusions are as follows:

(1) Speed: Replace the feature extraction network with the lightweight MobileNetV1 network with a model size of only 46.7 MB.

(2) Accuracy: Due to the small parameters of lightweight network, the detection accuracy decreased. Therefore, the attention mechanism is added, which can ensure the accuracy of detection while maintaining lightweight. Through this operation, the accuracy of the model on the test set is 87.5% for category <60% tomatoes, 88.9% for 61%_70% tomatoes, 81.1% for 71%_80% tomatoes, and the accuracy of tomatoes in the 81%_100% category is 93.4%. The accuracy of tomato maturity classification is above 80%.

(3) Compared with the traditional target detection, the proposed model is robust and can accurately detect tomatoes and judge the maturity of tomatoes under the condition that branches, leaves, and fruits are shielded from each other. Compared with the deep learning model detection research, the proposed model has advantages in reasoning speed, which is of great significance to the automatic picking and embedded research of tomato picking robot.

**Author Contributions:** Y.Z., G.W. and L.Z. collected data on greenhouse tomato fruits; Y.Z., F.S. and Y.Y. analyzed the data; F.S., Y.Z. and P.L. wrote the paper; F.S. and G.W. drew pictures for this paper; L.Z., Y.Y., P.L. and L.Z. reviewed and edited the paper. All authors have read and agreed to the published version of the manuscript.

**Funding:** This research was supported by the Shandong modern agricultural technology system (SDAIT-18-06), and the major agricultural applied technology innovation project of Shandong province (SD2019ZZ019).

**Institutional Review Board Statement:** Not applicable.

**Informed Consent Statement:** Not applicable.

**Data Availability Statement:** Not applicable.

**Acknowledgments:** The authors would like to thank all the reviewers who participated in the review.

**Conflicts of Interest:** The authors declare no conflict of interest.

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
