# Peer review of "Tomato Maturity Classification Based on SE-YOLOv3-MobileNetV1 Network under Nature Greenhouse Environment"

_agronomy, doi:10.3390/agronomy12071638_

Round 1

Reviewer 1 Report

Dear Authors, here are the detailed notes on the manuscript:

1) Line 32 - use a space between the word and the quotation [x] (use throughout the manuscript)

2) The purpose of the work has not been defined, correct it based on the content from the introduction; lines 74-78

3) What variety of tomato was used in the experiment? In my opinion, you confuse variety with species (line 106, fig. 3)

4) Equation # 3 - improve the appearance of the loss function

5) Figure 9. Test results of…. - do these values differ significantly? The drawing would be more reliable if error bars were inserted (5%, Sd or the average value - up to the Authors' decision)

6) Chapter 3.4. Analysis of test results - was the analysis based on any statistical tests?

7) I recommend extending your references with publications in the field of image analysis

Author Response

Thanks for your comments on our manuscript. Those comments are all valuable and very helpful for revising and improving our paper. We have studied the comments carefully and made modification on the original manuscript, which we hope meet with your approval.

Reviewer 2 Report

In the article presented classification of tomato maturity during it's production in the nature greenhouse environment. Heaving in mind the total paths of tomato production from plant to the table, it is very  important process to be considered. Described model is able to distinguish tomatoes in four kinds of maturity with precision equal 97,5 %. Such finding gives good possibility for evaluation of tomato colour sorter machine.

Please add new section number 4. Discussion. Then move Conclusions to section 5. Please add to section Conclusions subpoints, with the most important findings presented in the article.

Author Response

Thanks for your comments on our manuscript. We have studied your comments carefully and made modifications, hope our revised paper will meet your suggestions!
